# MOMA: Multi-Object Multi-Actor Activity Parsing

**Zelun Luo,**[*] **Wanze Xie,**[*] **Siddharth Kapoor,** **Yiyun Liang,** **Michael Cooper,**
**Juan Carlos Niebles,** **Ehsan Adeli,** **Li Fei-Fei**

Stanford University

{alanzluo, wanzexie, siddkap, isaliang, coopermj, jniebles, eadeli, feifeili}@stanford.edu
https://moma.stanford.edu

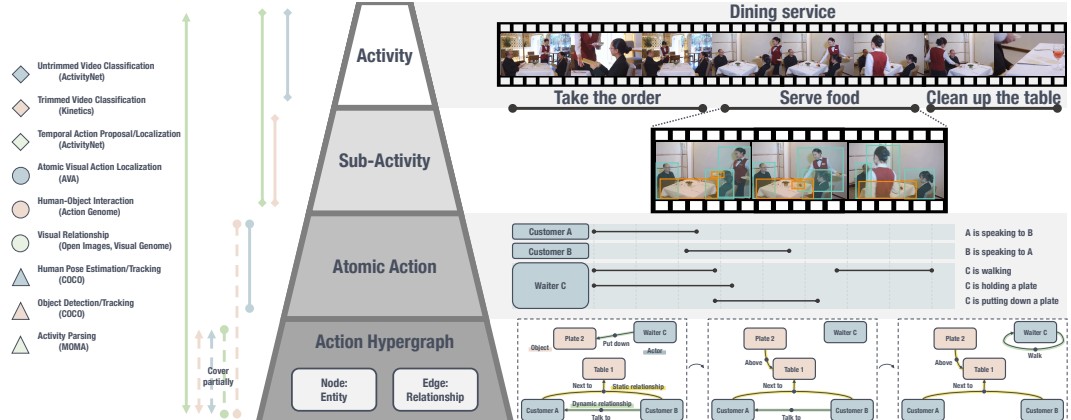

Figure 1: An overview of our paper. We introduce *Activity Parsing*, a partonomy-based activity recognition framework that considers the hierarchy and composition of complex human activities.

## Abstract

Complex activities often involve multiple humans utilizing different objects to complete actions (e.g., in healthcare settings, physicians, nurses, and patients interact with each other and various medical devices). Recognizing activities poses a challenge that requires a detailed understanding of actors' roles, objects' affordances, and their associated relationships. Furthermore, these purposeful activities comprise multiple achievable steps, including sub-activities and atomic actions, which jointly define a hierarchy of action parts. This paper introduces *Activity Parsing* as the overarching task of temporal segmentation and classification of activities, sub-activities, atomic actions, along with an instance-level understanding of actors, objects, and their relationships in videos. Involving multiple entities (actors and objects), we argue that traditional pair-wise relationships, often used in scene or action graphs, do not appropriately represent the dynamics between them. Hence, we introduce *Action Hypergraph*, a new representation of spatial-temporal graphs containing hyperedges (i.e., edges with higher-order relationships). In addition, we introduce *Multi-Object Multi-Actor (MOMA)*, the first benchmark and dataset dedicated to activity parsing. Lastly, to parse a video, we propose the *HyperGraph Activity Parsing (HGAP)* network, which outperforms several baselines, including those based on regular graphs and raw video data.

---

[*]These authors contributed equally.

35th Conference on Neural Information Processing Systems (NeurIPS 2021).

# 1 Introduction

While human activity recognition has advanced considerably in recent years, the datasets [8, 14, 68] and models [69, 74, 17] that underlie these improvements still treat human activities as monolithic events. Typically, the action classification task is only conducted on a coarse temporal scale and on a limited set of activity labels. In contrast, real-world activities in domains such as healthcare [87, 47, 53], surveillance [52, 78, 29], and entertainment [77, 57, 70] are complex, which involve intricate social interactions between humans utilizing different objects and sequences of steps they perform to achieve a goal. The understanding of these activities requires more fine-grained and structured reasoning beyond single semantic labels. Recently, efforts have been made in developing activity recognition models to address this disparity, including temporal localization [6, 95], hierarchical reasoning [65], atomic action reasoning [22, 21], and action decomposition [27]. However, none of them fully addresses the needs in real-world applications.

We argue that a complex activity recognition system invokes understanding the detailed *hierarchy* and *composition* of activity components in a video sequence [89]. Firstly, human activities are by nature hierarchical [3] and abstract behaviors (such as `dining service`) are composed of a series of low-level body movements and interactions (such as `speaking`, `sitting`, and `holding`). Defining a vocabulary of activity labels can be ambiguous if multiple levels of granularity (i.e., a hierarchy) are not adequately defined. Secondly, when multiple entities exist in the video, understanding the composition of the video by reasoning each actor's social role [61], each object's affordance, and their relationships [27] is crucial to visual dynamics understanding (see Fig. 1).

Towards this goal, we propose a redefined *Action Parsing* [39, 43, 66] for complex human activity recognition. Our proposed activity parsing task parses complex activities into multiple timescales and spatial-temporal grounding of entities and relationships. The representation consists of four levels of hierarchy: (1) activity, (2) sub-activity, (3) atomic action, and (4) action hypergraph. More specifically, an action hypergraph is a spatial-temporal graph [50] tailored for multi-object and multi-actor scenarios, in which a hyperedge can encode higher-order relationships by connecting any number of vertices (i.e., entities, including actors and objects). Hyperedges, which can still trivially represent pairwise relationships, define more comprehensive relationships than the pairwise relations prevalent in the traditional scene graphs. In this representation, the contextual information in high-level activities and sub-activities facilitates the fine-grained recognition of scene dynamics on the atomic action and action hypergraph levels; low-level action hypergraphs and their constituents provide a grounding for high-level activity recognition necessary for its interpretability and transferability.

Furthermore, we propose a novel model, termed Hypergraph Activity Parsing (HGAP) network, for the activity parsing task on videos. We also explore other tasks such as video classification and role classification with HGAP. We compare HGAP with an existing graph neural network approach [83] and the state-of-the-art activity recognition model [16]. By learning with Action Hypergraph, HGAP achieves comparable results with 3D CNN models without pre-training on videos from the same domain, and HGAP with X3D backbone can achieve superior results for activity parsing.

Lastly, we introduce the *MOMA (Multi-Object, Multi-Actor)* dataset, consisting of 17 activity categories, 67 sub-activity categories, 52 atomic action categories, 120 object categories, 20 actor categories, and 75 relationship (static and dynamic) categories. The MOMA dataset has three unique features: (1) activity annotations with four levels of granularity, which enable activity parsing on different levels of abstraction; (2) multi-object, multi-actor, and categorical labels for actors (i.e., social roles) and objects, which provide exhaustive details of the associated activity; (3) higher-order relationships [12, 1] between entities, which expressively capture the relationships in crowded and complex scenes. To the best of our knowledge, MOMA is the first dataset with these features.

# 2 Related work

**Action recognition** refers to the problem of recognizing human actions in videos, with two major tasks in the domain (1) action classification [91] and (2) temporal action detection (TAD) [79]. For (1), recent advances are toward 3D CNNs [8, 23, 60, 73, 80] such as C3D [73], I3D [8], SlowFast [17], and X3D [16] that jointly model spatial and temporal features. (2) [79] in comparison is a harder task, and works often follow either a proposal-then-classification paradigm [67, 81, 19, 42, 40] or an approach to model the two steps at same time [41, 5, 88, 2].

Table 1: A comparison of MOMA with related video datasets. MOMA, the first multi-object, multi-actor activity recognition dataset, offers rich annotations at four levels, with actors and objects represented as a novel action hypergraph.

| Dataset | Action partonomy | # videos | # hours | # action categories | Objects | | Actors | | Relationships | | |
|---|---|---|---|---|---|---|---|---|---|---|---|
| | | | | | localized | # classes | annotated | # classes | annotated | edge type | # classes |
| ActivityNet [14] | | 28K | 849 | 200 | | - | | - | | | - |
| Kinetics-700 [7] | | 650K | 1.8K | 700 | | - | | - | | | - |
| HACS Clips [94] | | 0.4K | 0.8K | 200 | | - | | - | | | - |
| Social Roles [61] | | 0.2K | 4.7 | 4 | | - | ✓ | 17 | | | - |
| Something-SomethingV2 [21] | | 220K | - | 174 | | - | | - | | | - |
| HVU [11] | | 572K | - | 739 | | 1.7K | | - | | | - |
| LEMMA [28] | | 324 | 43 | 641 | | 64 | | - | | | - |
| TITAN [48] | | 700 | 3 | 50 | ✓ | 2 | ✓ | 3 | | | - |
| AVA-Kinetics [36] | atomic action | 230K | - | 80 | | - | | - | | | - |
| Action Genome [27] | action | 10K | 82 | 157 | | | | | | | |
| | scene graph | - | - | - | ✓ | 35 | | - | ✓ | pairwise | 25 |
| FineGym [65] | action | 181 | 63 | 4 | | - | | - | | | - |
| | sub-action | - | - | 288 | | - | | - | | | |
| **MOMA** | activity | 373 | 66 | 17 | | - | | - | | | |
| | sub-activity | 2.4K | 10 | 67 | | - | | - | | | |
| | atomic action | 12K | 10 | 52 | | - | | | | | |
| | action hypergraph | - | - | - | ✓ | 120 | ✓ | 20 | ✓ | hyperedge | 75 |

Today, researchers have pushed human action understanding to a deeper level in two major directions that our work is based upon: (1) action as *hierarchical representations*, (2) action as compositions of *visual relationships*. To jointly model these two representation, we identify *graph neural network* and *action parsing* as two promising tools for our study, briefly surveyed below.

**Hierarchical representation learning** is often studied on data with a network structure [4, 90, 9, 51], yet its application in video action recognition is an emerging direction. Drawing from Cognitive Science studies [89], a hierarchy of human actions can either be *taxonomic* [14, 11] (e.g., `playing basketball` is a kind of `sport`), or *partonomic* [33, 35, 27, 65] (e.g., `take an order` is a step of `dining service`). [45, 72] leveraged taxonomy hierarchy for action prediction, whereas [35, 33, 34] exploited the two-level partonomy of actions for few-shot action classification. This work aims to apply a four-level partonomic hierarchy to solve the task of activity parsing. In particular, we break high-level activity to sub-activity, then atomic action, and lastly a frame-wise representation for visual human-object relationship, as discussed in the next subsection.

**Visual relationship and compositionality** has been studied in video domain to represent human actions. For example, Something-Else [49] models object relationship as tuples, Action Genome [27] formulates human-object relationship in video as a spatial-temporal scenegraph, and STAG [25] views changing relationship as dynamic graph representing an action. To model this, models [46, 82, 38] and new loss functions [93] are proposed. In recent years, graph neural networks [85, 56] have been found to be a promising tool for both generating and learning from visual compositions.

**Graph neural networks (GNNs)** were first introduced in [20, 64], and has been recently adopted for video understanding. Lines of work have studied its application on spatial-temporal graph representations such as skeleton data [84, 37] or visual relationships [59, 54]. More recently, Hypergraph Neural Network [18] was proposed and greatly inspired our work as we see its potential in improving scene graph representations and deepen model's cognitive understanding to human actions.

**Action parsing** [66, 15, 43, 44, 39, 58, 76, 55, 75] is often referred to as modeling multiple characteristics of actions simultaneously. For example, [39] jointly models human joints, objects, and action labels in images, whereas [44] explored predicting human pose, action attributes and action labels together in videos. In this work, we redefine this task to be *activity parsing*. We will ask the model to predict the activity, to temporally detect the sub-actions that occurred, to localize the people and the objects that involved, and to recognize the visual relationships related to the ongoing activity.

## 3 Data representation and dataset

The MOMA dataset is structured in a four-level hierarchy based on the activity partonomy, with rich annotations at each level. As shown in Figure 1, the *activity* `dining service` consists of multiple steps (i.e., *sub-activities*), including `take an order`, `serve wine`, `serve food`, and `clean up the table`. Further, the sub-activity `taking an order` also involves multiple entities. It consists of several actor-centric *atomic actions*: `sit`, `talk to someone`, `write on something`, etc. At the lowest level, we provide a dense annotation of actor-object relationships. In particular, we annotate *higher order relationships*, such as (\{waiter\}, `holding`, \{pen, sheet\}), (\{customer,

waiter}, on the side of, {dining table}). We argue that its outcome, the *action hypergraph* is more effective when representing multi-object multi-actor interaction. In addition, we provide temporal localization for each sub-level action, and assign a unique ID for each actor/object in a video for temporal tracking.

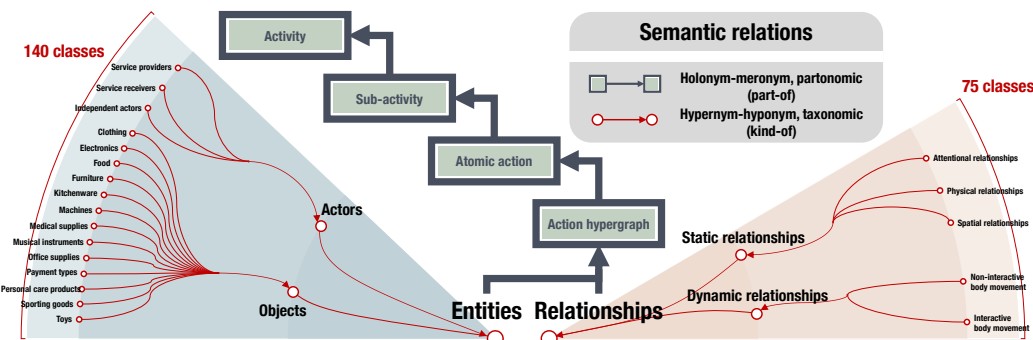

Figure 2: A conceptual diagram of the semantic relations introduced in this paper, which includes partonomic and taxonomic hierarchies. The partonomic hierarchy is used for activity parsing, and the taxonomic hierarchy is used for categorizing entities and relationships.

## 3.1 Dataset statistics

MOMA contains 373 raw videos at activity level, 2,364 trimmed videos at sub-activity level, and 12,057 atomic action instances. This includes 17 activity classes, 67 sub-activity classes, and 52 atomic action classes. At frame level, we provide action hypergraph annotations for $37,428$ frames, with $164,162$ actor/object instances of 20 actor classes and 120 object classes, and $119,132$ relationship instances of 75 relationship classes. On average, there are $4.39$ actors/objects and $3.18$ higher-order relationships per frame, 5 instances of atomic actions per clip, and $6.34$ instances of sub-activities per untrimmed video (Table 1). Detailed information is provided in the supplementary material about the distributions of activities, subactivities, atomic activities, entities, relationships, and their co-occurrences.

We have adopted a series of protocols to mitigate potential ethical issues of the dataset: (1) videos are sourced from the YouTube platform, where each video has been cross-verified by multiple annotators to avoid offensive contents; (2) we do not provide videos, but instead provide YouTube IDs and download script; (3) we control personally identifiable data so that only links to publicly available videos are released, and owners of the video can disable the accessibility at any time; (4) any release of imaging data for demonstration purpose will be verified by our group to not contain any personally identifiable information, and will involve using face blurring algorithms.

## 3.2 Partonomic hierarchy

The MOMA Representation features a four-level partonomic representation for complex activities.

**Activity.** An activity [69] (e.g., `dining service`) is a high-level description of an event done by people for a specific purpose. Typically, it is an abstraction of several sub-processes carried out by multiple entities. Spurious features, such as background, often provide contextual clues important for sub-optimal activity classification.

**Sub-activity.** A sub-activity is a stage making up part of a larger activity. Typically, the sub-activities (e.g., `serve food`, `clean up the table`) from the same activity may take place within the same environment. Hence, the key to segmenting and classifying a sub-activity is to identify actors, objects, and their interactions.

**Atomic action.** An atomic action [22] represents an actor's body movement or interaction with other entities. At the most primitive level, atomic actions are actor-centric [71], but an actor can simultaneously perform multiple atomic actions. Generally speaking, atomic actions must be spatially and temporally localized in order to be recognized.

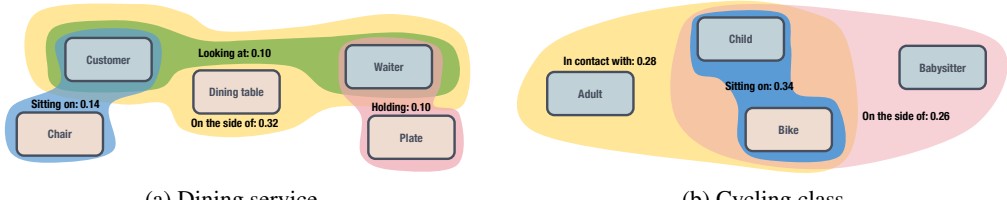

(a) Dining service.          (b) Cycling class.

Figure 3: Average action hypergraphs for `dining service` (Figure 3a) and `cycling class` (Figure 3b). We average all action hypergraphs in an activity class and select the most frequent nodes and edges. A larger hyperedge value indicates a higher frequency within the associated activity.

**Action hypergraph.** An action hypergraph is a spatial-temporal heterogeneous hypergraph that incorporates static and dynamic information about actors and objects. In an action hypergraph, a node represents an entity (actor or object) and a hyperedge (which can join more than two nodes) represents a higher-order relationship (static relationship or dynamic relationship) between the connected entities. As a spatial-temporal graph, an action hypergraph changes with time [50] and its nodes are spatially localized and tracked as bounding box tracklets on the raw video. As a result, although the action hypergraph is annotated per frame, based on the granularity of our prediction, it is flexible to be formed into a hypergraph that captures information about higher level actions (e.g. atomic actions, sub-activities, or activities).

## 3.3 Details of action hypergraph

Action hypergraph is a novel spatial-temporal representation at the bottom level of the MOMA hierarchy. It features: (1) higher-order relationships that connect multiple entities with hyperedges; (2) class labels for both actors (social roles) and objects (object categories); (3) temporal tracking for entities, resulting in spatial-temporal graphs.

**Entity.** We define two types of entities, namely *actor* and *object*. In an action hypergraph, each entity is represented as a node and is defined spatial-temporally (bounding box tracklet) and categorically (actor role or object category). Using a taxonomic hierarchy, we distinguish three social roles (*service provider*, *service receiver*, and *independent actor*) within 20 actor classes and 13 object categories (*kitchenware*, *clothing*, *medical supplies*, etc.) within 120 object classes. We illustrate the taxonomic hierarchy of entities in Figure 2 and provide the complete taxonomy in the supplementary material.

**Relationship.** We define two types of relationships, namely *dynamic relationship* and *static relationship*. A dynamic relationship shows a continued or progressive action initiated by an actor and it usually lasts for a few seconds. On the contrary, a static relationship describes a state of being among two or more entities and its duration is indefinite. In an action hypergraph, each relationship is represented as an edge and is defined categorically. Unlike previous studies [32, 27, 13], we introduce higher-order relationships using hyperedges. The hyperedge provides a versatile way to represent a single actor's body movement (as a self-loop), a relationship between two entities (as a regular edge, i.e., a trivial hyperedge), or a relationship between more than two entities (as a non-trivial hyperedge). Using a taxonomic hierarchy, we differentiate *non-interactive* (associated with just one entity) and *interactive* (associated with more than one entity) movements within 52 dynamic relationship classes. Similarly, we distinguish *spatial*, *physical*, and *attentional* relationships within 23 static relationship classes. We illustrate the taxonomic hierarchy of relationships in Figure 2 and provide the complete taxonomy in the supplementary material.

**Significance of higher-order relationships.** Mathematically, a regular graph can be converted to a hypergraph by merging edges that have same attributes and come from same nodes. However, this is not the case for MOMA dataset. In human interactions, higher-order relationships contain strictly more information than its pair-wise equivalents. For example, during annotation, (`waiter`, `looking at`, `glass`), (`waiter`, `pouring water into`, `glass`), is likely to be separated from (`two customers`, `looking at`, `glass`). Here, different instances of `looking at` are not merged because they serve different purposes and do not share a same intention from the actors. More examples are shown in the supplementary section.

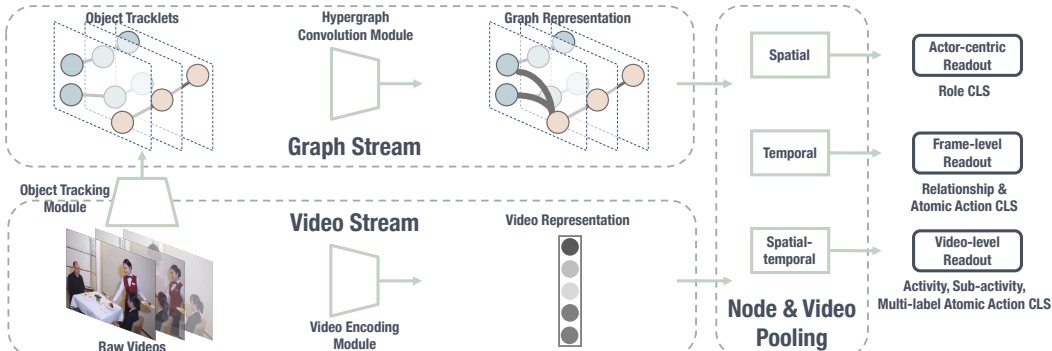

Figure 4: The structure of the HGAP model. In our activity parsing tasks, we design the multi-head MLP with different type of output heads for different prediction result, which is covered in detail in Section 4.2. *CLS* stands for classification.

# 4 Method

We introduce the HyperGraph Action Parsing (HGAP) network, which uses the action hypergraph representation as a foundation for activity parsing, hierarchical action classification, temporal segmentation, and actor role classification (Figure 4).

## 4.1 Action hypergraph representation

The action hypergraph of a video of length $n$ is defined as $G = (V, \mathcal{E}, R, T)$. $T = \{t^{(1)}, t^{(2)}, \cdots\}$ is the set of *node types*. In videos, a node type corresponds to an actor role or an object category. $R = \{r^{(1)}, r^{(2)}, \cdots\}$ is the set of *edge types*. An edge type corresponds to a relationship category. $V = \{v_1, v_2, \cdots\}$ is a set of *nodes*. A node correspond to a specific actor or object instance. $\mathcal{E} = [E_1, E_2, \cdots]$ is a length-$n$ list of sets, one for each frame. Each set $E_i = \{e_{i,1}, e_{i,2}, \cdots\}$ is a set of *directed hyperedges*, each of which $e_{i,j} = (V_{i,j}^{head}, r_j, V_{i,j}^{tail})$ is defined by a tuple of head nodes $V_{i,j}^{head} \subset V$, tail nodes $V_{i,j}^{tail} \subset V$, and a relationship type $r_j \in R$, where $i \in [1, n]$ and $j \in [1, |E_i|]$. A directed hyperedge corresponds to a high-order relationships between two sets of entities in a specific frame.

An action hypergraph is a graph that is (1) a hypergraph (i.e., a hyperedge can join any number of nodes); (2) heterogeneous (i.e., contains different types of nodes); (3) dynamic (i.e., the graph changes with time). More specifically, in an action hypergraph, only the set of hyperedges evolve over time and the set of nodes remains constant. An actor or object that has disappeared or has not yet appeared in the scene have the degree $0$.

## 4.2 HGAP network

The HGAP Network is a two stream network that combines a graph branch for modeling action hypergraph, and a video branch that serves as a backbone to extract the RGB features across the video. In this section, we describe our implementation based for the graph branch.

Given a trimmed video $v = \{i^{(1)}, i^{(2)}, \ldots, i^{(N)}\}$ of $N$ frames, the model outputs a set of $n_a$ bounding box tracks of actors $T_a = \{t_a^{(1)}, t_a^{(2)} \ldots, t_a^{(n_a)}\}$, and $n_o$ bounding box tracks of object instances $T_o = \{t_o^{(1)}, t_o^{(2)} \ldots, t_o^{(n_a)}\}$. Each track $t = \{b^{(1)}, b^{(2)}, \ldots, {}^{(m)}\}$ consists of a sequence of bounding boxes. Note that we assume the availability of actor and object bounding boxes (but not the categories) because object detection [63, 62] has achieved impressive result and is beyond the scope of this paper.

**Hypergraph modeling.** To represent the hypergraphs, we propose to utilize hypergraph convolution borrowed from hypergraph neural networks (HGNN) [18]. Note that HGAP's graph branch can be replaced with any grahp-based feature extraction model . Hence, as a baseline model, we use Graph Convolutional Networks (GCN) [31] to compare our results with. Below we describe the setup of our general graph encoder and its readout functions.

**Nodes.** Nodes are defined as the entities (actors and objects) in a trimmed video. We further augment the node features with image features extracted from the bounding box regions, which can also be replaced with region proposal features from models (Faster R-CNN [63]) for end-to-end training.

**Edges.** Edges are defined as the relationships between entities in a trimmed video. In one of our tasks, the goal is to predict the category of edges in the graph.

**Readouts.** Readouts are defined with three strategies for different output heads. We first use MLP to map node features to a hidden dimension with the same encoder, and then we have actor-centric readout, frame-level readout, video-level readout, each designed for specific tasks.

(1) *Actor-centric readout* applies temporal average pooling for actor node's features across the entire video. This readout function is followed by role prediction heads for each actor across the video.

(2) *Frame-level readout* applies global average pooling across for all nodes in the graph. This readout connects to the per-frame classification heads for relationship classification and atomic action localization.

(3) *Video-level readout* first performs nodes pooling to generate embedding for each graph, and then applies temporal average pooling to gather the information of all graphs in the video. This readout is designed for all video-level tasks including activity classification, sub-activity classification, and multi-label atomic action classification.

**Hyperedge convolution layer.** In HGAP Network, we follow the original design [18] of hyperedge convolution operator and build the convolution layer $f(X, W, \Theta)$ as follows

$$X^{(l+1)} = \sigma(D_v^{-\frac{1}{2}} H W D_e^{-1} H^T D_v^{-\frac{1}{2}} X^{(l)} \Theta^{(l)}),$$

in which $X^{(l)} \in R^{N \times C}$ refers to an embedding matrix with $N$ nodes of $C$ feature channels, $D_v$, $D_e$ defines the vertex and edge degree matrices of the hypergraph, and $\Theta$ is the weight matrix in the update function. In our model, we use the bounding box info and the average pooled region of interest (RoI) as input feature for the actor and object nodes, with an equal weight $W$ for all hyperedges, and ReLU activation for $\sigma$.

We show that by leveraging the tracking information and creating a spatial-temporal hypergraph, we are able to capture more nuance in the actions with short duration and yield better outcome for hierarchical predictions.

### 4.3 Video feature extraction

The video branch of the HGAP network is built upon the state-of-the-art action recognition model X3D [16] shows promising performance on a variety of video action datasets including Charades [68], AVA [22], and Kinectics [30]. We use the original X3D_L as a baseline approach for hierarchical action classification, and compare it with HGAP's performance on same tasks.

The input to the video model includes a trimmed video sequence $v = \{i^{(1)}, i^{(2)}, \ldots, i^{(N)}\}$ of $N$ image frames with sample rate of $8$ with $N = 16$. We pretrain the X3D_L model on Kinetics-400 [30], and then modify the original X3D_L model with a multi-head output to enable 3-level hierarchical action classification for activity, sub-activity, and atomic actions. In particular, we perform multi-class single-label classification for activity and sub-activity level output, and perform multi-class multi-label classification for a set of atomic actions that show up in the entire trimmed video. We train this video branch independently, and pick the model with best validation result across the three tasks.

To incorporate the video feature extraction backbone to the HGAP network, we concatenate the output heads of X3D with the video-level output heads of the graph branch, which include the activity classification head, sub-activity classification head, and multi-label atomic action classification head. At test time, we average the logits of the two branches together and generate the prediction output.

## 5 Experiments

In this section, we examine three different tasks using the MOMA dataset. The first task is activity parsing, which fully leverages the strength of our dataset. Second, we conduct experiments on the hierarchical classification task to compare HGAP with the state-of-the-art action classification model. Finally, we consider role classification, which is rarely discussed in existing work but critical to

Table 2: The performance of HGAP Network on activity parsing. "Oracle" indicates that we use ground truth graph structure (without class label for node and edge) as input to the network. For GCN is not capable of handling hyperedges, we show its result for pair-wise edge classification.

| | | | | | (All results are reported as mAP %) | | |
| Task Level → | Activity | Sub-activity | Atomic Action | | Action Hypergraph | | |
| Methods ↓ | Classification | Classification | Classification | Localization | Hyperedge Classification | Role Classification | Edge Classification |
|---|---|---|---|---|---|---|---|
| GCN [31] | 70 | 37.4 | 29.4 | 29.1 | - | 64.7 | 37.7 |
| HGAP | 73.9 | 42.5 | 29.2 | 30.3 | 27.6 | 46.4 | - |
| HGAP Oracle | 88.4 | 44.1 | 31.3 | 32.7 | 36.3 | 54.3 | - |

understanding video semantics. For all three experiments, we split `80%` of data for training, and hold out the rest for validation. The major hyperparameter we focus on is the loss weight assigned for each output head, while we also tune on model hidden size and initial learning rate. We use an 8-GPU (Tesla V100) environment for training video feature extractor and fine-tuning on the action hypergraph. We demonstrate the significance of action hypergraph representation through these three experiments and the carefully designed baselines.

## 5.1 Activity parsing

The purpose of the activity parsing task is to demonstrate that the model has a comprehensive cognitive understanding of an activity. Specifically, we train a model capable of detecting and classifying activities, sub-activities, atomic actions, as well as action hypergraphs within the video. In addition to raw video data, we assume bounding box tracklets of people and objects in the scene are available. This paper will not concentrate on object detection and tracking since other studies [63, 86, 92] have demonstrated excellent performance. Instead, the paper focuses specifically on temporal reasoning. We also perform an *Oracle* experiment with ground-truth structures of action hypergraphs (without knowing the classes of nodes and edges). The results show that an accurate action hypergraph is crucial for the overall activity parsing performance.

**Problem formulation.** Activity parsing is defined as a joint prediction of the following characteristics: (1) activity category; (2) sub-activity category; (3) atomic action categories with temporal localization; (4) an action hypergraph comprised of all nodes' and edges' categories. We define inputs to be (i) a trimmed video with a single sub-activity that is segmented from the complete video, and (ii) tracklets that represent actors and objects across the trim video. Tracklets are defined as temporally stacked RoI for the actor or object in the video.

**Evaluation metrics.** We find that mean average precision (mAP) [26] is a good metric for all sub-tasks defined above. It is suitable for multi-class classification when sampling is spread out over a long tail distribution like in our case. For the classification of atomic actions and actor roles, multi-label mAP [68] is used. We also adopted it for the task of localizing temporal atomic actions, because atomic actions are highly overlapped both spatially and temporally, as multiple people can perform different atomic actions at the same time. Thus, each frame is predicted with a set of atomic actions, and mAP is computed for each frame at granularity of one frame per second.

**Baselines.** Since activity parsing is a newly proposed task, there are no existing published baselines. However, our objective is to demonstrate the utility of the action hypergraph representation, and our baseline is therefore modified from the Graph Convolutional Network (`GCN`) model [31] to take in RoI as node feature, and train with multiple output heads, where hyperedges are decomposed to pair-wise ones.

**Results.** Table 2 reports activity parsing experiment performance. `HGAP` achieves better or comparable performance compared to `GCN` on all high-level tasks including classification for activity, sub-activity and atomic actions, along with atomic action temporal localization. Interestingly, `HGAP` performs better on all high level tasks even though `GCN` performs better on graph level classifications. We have seen evidence in the image and video domain that the learning of visual relationships increases performance on high-level tasks such as image captioning [32], collection detection [25], and action recognition [13, 27]. This thus illustrates that hypergraphs are intrinsically harder to learn, but their dense feature representation enables the model to better understand the video semantics, even if the hypergraph itself is not sufficiently learned. `HGAP Oracle` shows that greater performance gains with a better hypergraph model.

Table 3: The performance of HGAP network on the hierarchical activity classification task.

| Method | Pre-train | (All results are reported as mAP %) | | |
|---|---|---|---|---|
| | | Activity | Sub-activity | Atomic Action |
| X3D [16] | Kinetics-400 [30] | 70.9 | 40.6 | 27.1 |
| X3D + GCN | Kinetics-400 | 71.8 | 41.9 | 29.7 |
| HGAP | Kinetics-400 | 73.9 | 42.5 | 29.2 |
| HGAP Oracle | Kinetics-400 | 88.4 | 44.1 | 31.3 |

Table 4: The performance of HGAP network on role classification in comparison with ResNet-50.

| Method | Pre-train | mAP % |
|---|---|---|
| ResNet-50 [24] | ImageNet [10] | 37.9 |
| HGAP | Kinetics-400 | 46.4 |
| HGAP Oracle | Kinetics-400 | 54.3 |

## 5.2 Hierarchical action classification

In this experiment, we further evaluate the utility of the action hypergraph representation. Since the proposed HGAP combines video information with learned action hypergraphs, we hope to show that HGAP achieves a performance improvement on the action classification task, even when compared with the state-of-the-art action model augmented with a regular graph.

**Problem formulation.** To fully assess the model's capability to understand ongoing activity, we evaluate a joint multiclass classification across three action levels. We use the same inputs as in activity parsing, but we predict (1) activity categories (single-label, multi-class), (2) sub-activity categories (single-label, multi-class), and (3) atomic action categories (multi-label, multi-class). mAP is used as the metric for sub-tasks (1) and (2), and multi-label mAP for sub-task (3).

**Baselines.** We implemented two baselines. The first baseline is a X3D_L model [16] that shows state-of-the-art performance on several challenging action datasets, but customized with a multi-headed output layer that incorporates RGB video along with the flattened graph matrix. The second is a X3D_L model enhanced with GCN, which allows it to learn structured graph features from the action hypergraph (with hyperedges broken to pair-wise ones).

**Results.** As shown in Table 3, HGAP outperforms X3D on all three sub-tasks, and achieves a better average performance than X3D combined with GCN, with a 2.1% mAP margin on the activity classification sub-task. While HGAP Oracle performs even better, it is important to note that this is not an upper bound, not only because we did not use ground-truth actor or object labels, but also because ground-truth graph structure is sampled at a rate of 1 second per frame.

## 5.3 Role classification

In video datasets, role classification is rarely defined except briefly touched upon in [61, 28]. Using dense annotations and comprehensive actor role categories in MOMA, we aim to highlight this problem as a stand-alone task and promote a research direction to the joint modeling of actor-centric actions and actor roles within ongoing activity.

**Problem formulation.** In this experiment, we investigate how understanding visual relationships in the on-going activity enhances the model's ability to recognize each actor's role in the video. The inputs are identical to those in activity parsing, but the output is a multi-label, multi-class role prediction for all the people in the video. Our prediction is based on the whole actor tracklet instead of a single frame, by first aggregating all RoIs with the temporal tracking ID, and then classifying the actor's role. mAP is used as an evaluation metric for this task.

**Baseline.** The baseline for this evaluation is a classifier based on ResNet-50, which accepts tracklets as input. In this algorithm, tracklets are processed on a frame-level to produce role class distributions for each frame in each tracklet, and then the average of these distributions across frames is used to predict actor roles for each tracklet. To emphasize the importance of visual relationships, this baseline does not use graph structure on purpose.

**Results.** Table 4 compares the baseline model to two proposed models, with HGAP outperforming ResNet-50 by 8.5%. Further, HGAP Oracle can improve mAP scores by 7.9% when the ground-truth graph structure is applied. The results indicate that learning a good action hypergraph representation is indeed useful for role classification, and we can expect that a more advanced hypergraph modeling technique will result in even greater improvements.

# 6 Conclusion

We created a first-of-its-kind video dataset, MOMA, that (1) identifies a new list of activities involving multi-person collaboration, (2) formulates a four-level hierarchical representation of activity, and (3) proposes Action Hypergraph, a spatial-temporal heterogeneous hypergraph representation for visual relationships. Furthermore, We defined activity parsing, a new task to show a model's holistic understanding of an activity, and designed `HGAP`, a new architecture that combines video feature extraction with hypergraph neural networks and achieves superior result on a variety of tasks. We anticipate that MOMA opens up a wide range of potential future research directions, and helps future works toward better action recognition and with providing the means for holistic and detailed understanding of the videos. Detailed and structured understanding of activities and their components (i.e., Activity Parsing) as offered by MOMA, when combined with large-scale unlabeled data, may facilitate few-shot or even zero-shot action recognition.

## Acknowledgment

This work was partially supported by the Schmidt Futures, Toyota Research Institute (TRI), and National Science Foundation grant 2026498. This study also benefited from Stanford Institute for Human-Centered AI (HAI) AWS Cloud Credits.

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

## 7.1 For all authors...

**(a) Do the main claims made in the abstract and introduction accurately reflect the paper's contributions and scope?**

Yes. Paper's contribution is summarized in abstract, and reflected in further detail in the last 3 paragraph of the introduction section.

**(b) Have you read the ethics review guidelines and ensured that your paper conforms to them?**

Yes, we can confirm.

**(c) Did you discuss any potential negative societal impacts of your work?**

Yes, we have touched upon the society impact of our work in both ways (positive application: healthcare, negative application: surveillance), as highlighted in the first paragraph of the introduction. Regulations and transparency of work's application are deemed as necessary to mitigate such impact.

**(d) Did you describe the limitations of your work?**

Yes, as highlighted in result in section 5 of the paper. The proposed model has limited capability in predicting hypergraph, yet we show how such limitation in turn demonstrates the effectiveness of the hypergraph representation for supporting high-level tasks.

## 7.2 If you are including theoretical results...

**(a) Did you state the full set of assumptions of all theoretical results?**

n/a

**(b) Did you include complete proofs of all theoretical results?**

n/a

## 7.3 If you ran experiments...

**(a) Did you include the code, data, and instructions needed to reproduce the main experimental results (either in the supplemental material or as a URL)?**

Yes. Steps to reproduce experimental results are covered in the supplemental material. Code, data, and further instructions will be released at `https://moma.stanford.edu/`.

**(b) Did you specify all the training details (e.g., data splits, hyperparameters, how they were chosen)?**

Yes. They have been summarized in first paragraph of section 5, and explained in detail in the Problem Formulation and Baseline section.

**(c) Did you report error bars (e.g., with respect to the random seed after running experiments multiple times)?**

No, we don't report error bars, confidence intervals, or statistical significance tests as they are irrelevant to supporting our main claim.

**(d) Did you include the amount of compute and the type of resources used (e.g., type of GPUs, internal cluster, or cloud provider)?**

Yes, we have specified them in first paragraph of section 5.

### 7.4 If you are using existing assets (e.g., code, data, models) or curating/releasing new assets...

**(a) If your work uses existing assets, did you cite the creators?**

n/a. The proposed asset is one of the major contributions in our paper.

**(b) Did you mention the license of the assets?**

n/a.

**(c) Did you include any new assets either in the supplemental material or as a URL?**

Yes, the proposed MOMA dataset will be released at `https://moma.stanford.edu/`.

**(d) Did you discuss whether and how consent was obtained from people whose data you're using/curating?**

n/a. We do not collect data from any particular individual, and we only provide access to the publicly available data, whose owner can choose to disable accessibility at any time.

**(e) Did you discuss whether the data you are using/curating contains personally identifiable information or offensive content?**

Yes, as it has been covered in detail in section 3.1 of the paper.

### 7.5 If you used crowdsourcing or conducted research with human subjects...

**(a) Did you include the full text of instructions given to participants and screenshots, if applicable?**

n/a

**(b) Did you describe any potential participant risks, with links to Institutional Review Board (IRB) approvals, if applicable?**

n/a

**(c) Did you include the estimated hourly wage paid to participants and the total amount spent on participant compensation?**

n/a

