# MOMA: Multi-Object Multi-Actor Activity Parsing Appendix

**Zelun Luo**,[*]  **Wanze Xie**,[*]  **Siddharth Kapoor**,  **Yiyun Liang**,  **Michael Cooper**,
**Juan Carlos Niebles**,  **Ehsan Adeli**,  **Li Fei-Fei**

Stanford University

{alanzluo, wanzexie, siddkap, isaliang, coopermj, jniebles, eadeli, feifeili}@stanford.edu
https://moma.stanford.edu

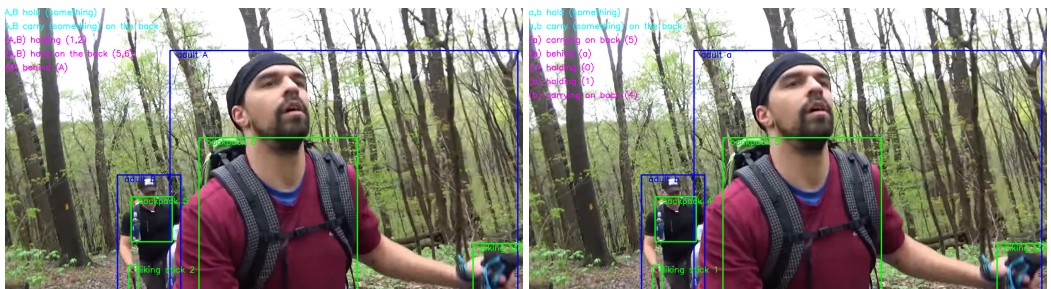

Figure 1: An example of errors that our checking algorithm can automatic identify. Left figure shows an example from previous iteration of the annotation, in which our algorithm automatically detects the ambiguity in higher order relationships of "holding" and "have on the back", since it is rare for two people to hold or carry two same things during hiking at same time. Right figure shows the current annotation, which ensures that ambiguous higher order relationships are replaced with the correct ones.

## A1   MOMA Data Curation

The MOMA dataset is comprehensive and covers many aspects of human activities. In summary, our annotations include following 10 different aspects.

- Activity: (1) activity label;
- Sub-activity: (2) sub-activity label, (3) temporal localization of sub-activity;
- Atomic action label: (4) atomic action label, (5) spatial and temporal localization of atomic action;
- Action hypergraph: (6) actor and object labels, (7) spatial localization of actor and object, (8) cross reference for actor and object in video via temporal tracking id, (9) relationship label, (10) actors and objects involved in each relationship.

With MOMA's comprehensive annotations, we strive to draw a clear definition for each category we annotate. The canonical definitions for each class and hierarchy have been given in Section 3 of the main paper. In the following subsections, we first present the annotation strategies we deployed to achieve the comprehensive labels mentioned above, then show the full taxonomy of all defined class labels, and lastly explain the tools we used to ensure the quality of our annotation given the complexity of the MOMA dataset.

---

[*]These authors contributed equally.

35th Conference on Neural Information Processing Systems (NeurIPS 2021).

**A1.1 Annotation procedure**

**Video sourcing for activity.** Based on common social interactions in daily life, we curated a target list of activities and sub-activities that may involve multiple entities. Then, given set of keywords related to each activity, we searched videos in batch from YouTube. We skimmed through each video, decided whether it can be categorized under one of our activity class, then we pre-defined a list of actors, objects, relationships, and atomic actions.

**Sub-activity annotation.** In this step, we annotated the temporal coordinates as well as the class label of the sub-activities that occur in the collected video. To scale this process, we developed a proprietary cloud-based web application, where multiple annotators can log in and get assigned with unprocessed videos. After this process, we trimmed raw videos into clips corresponding to a sub-activity class.

**Atomic action and relationship annotation.** At a sample rate of 1 fps, we converted trimmed videos into sequences of frames. During annotation, we provided sub-activity label as a hint and ask annotator to identify atomic actions in each frame similar to AVA [4], and assign each atomic action to a specific actor. The annotators then created bounding boxes for actors and objects related to the action, assigned the role/object label, and a unique ID for it across the video.

In the last step, annotators assigned relationship labels to a set of actors/objects. When there are multiple identical relationships occur simultaneously, we gave annotator the freedom to either annotate it as pair-wise relationship such as "$(a)$, *holding*, $(1)$", or higher-order relationship such as "$(a, b, c)$, *looking at*, $(1)$". We explain in further detail in A2.4 about why this annotation process yields more information for multi-object multi-actor relationships.

**Dataset cleaning.** We allowed annotators to create new class labels if they identify new objects, relationships, or atomic actions relevant to the given sub-activity. After one iteration of annotation, we analyzed the frequency of each label and decided strategies for merging or dropping labels.

We also designed a series of self-checking algorithms based on a set of rules for each type of label. For example, we ensure the existence of corresponding objects and actors for the annotated relationships and atomic actions in each frame, and identify possible ambiguity in higher order relationship annotations. A comprehensive list of our checking algorithm is included in section A1.3.

**A1.2 Class taxonomy and canonicalization**

In addition to the definition of actor, object, and relationship we have given in section 3 of the main paper, we also categorize these 3 set of classes into a 2-level taxonomy based on certain criteria. We describe the criteria below and present the full taxonomy.

- ACTOR: We categorize actor classes based on the properties of the roles in the scene, and separate them to "Service Provider", "Service Receiver" and "Independent". Table 1 shows the full taxonomy of actor classes.

- OBJECT: We categorize objects based on the topics of the objects itself. For example, piano and piano book are related in topic and are thus categorized into the same group. In total there are 12 groups of object categories. Table 5 shows full taxonomy of object classes.

- RELATIONSHIP: Notably, we improve from Action Genome [6]'s categorization of relationships, and give a clear definition for the three types of relationship including "attentional", "physical", and "spatial". The definition of the three types and their distinction from atomic actions are given section 3.3 of the main paper. Table 2 shows the full taxonomy of relationship classes. For convenience purpose, in this work, the word "relationship" specifically refers to "static relationship" unless otherwise emphasized, so as to be distinguished from the "dynamic relationship", as explained below.

- DYNAMIC RELATIONSHIP: Because action actions are essentially interactions between human and the environment (e.g.: someone wipes something), we believe it can be viewed as a type of relationship that is dynamically changing. When predicting higher-level actions (i.e. sub-activity or activity), we can merge this kind of relationship into the spatio-temporal action hypergraph to incorporate information from the atomic action level. Since atomic action verbs are represented as edges in the hypergraph, we denote the atomic action verb as "dynamic relationship", so as to be consistent with "static relationship" to formulate our

two edge types. As a result, the atomic action and dynamic relationship share the same verb vocabulary, but atomic action is the complete phrase (e.g. a person wipes table), whereas dynamic relationship is the exact verb (e.g. wipe).

## A1.3 Quality control

The annotation procedure has described above in detailed in section A1.1. As mentioned above, we employed a series of self-checking algorithm to identify the errors in action hypergraph annotations as well as to serve as regression test for any update to the annotation file. For every iteration of the annotation, we identify errors in annotation with the help of these algorithms until the annotations pass all the tests. These tests include:

1. SUB-ACTIVITY
   - Test if different sub-activities within one video overlaps with each other.
   - Test if all sub-activities are within time range of 3 sec - 30 sec.
   - Test if one video contains more then 10 instances of same class type of sub-activity

2. ACTOR&OBJECT
   - Test if the bounding boxes for objects and actors are rectangular.
   - Test if unique id is assigned for each object and actor.
   - Test if all actors are assigned with alphabetic ids, and all objects are assigned with numeric ids.

3. ATOMIC ACTION
   - Test if atomic action is attached to the corresponding actor.
   - Test if the actor id in atomic action is alphabetic (not numeric)
   - Test if the corresponding actor show up in the current frame.
   - Test if repetitive atomic actions are assigned to the same actor.

4. RELATIONSHIP
   - Test if all entities (actors and objects) in each relationship show up in the current frame.
   - Test if the relationship follows the definition described in section 3 of the main paper, and report any ambiguity found (Figure 1 shows an example of this test).
   - Test if each relationship connects at least two entities.
   - Test if newly created atomic action labels overlap with relationship labels.

Beyond leveraging the algorithmic tools to detect errors and have annotators resolve them, we also assign annotators to cross-verify annotations in each iteration, and we pay particular attention to:

1. SUB-ACTIVITY
   - Cut the rest of the video if one class type of sub-activity occurs more then 10 times within the video. This helps ensure the diversity of sub-activity instances across the MOMA dataset.

2. ACTOR&OBJECT
   - Select the most accurate description for each actor. For example, an actor should not be labeled as an adult if it can be classified as an athlete.
   - Select only actors and objects that are related to the ongoing action.

3. ATOMIC ACTION
   - Annotate only atomic actions that are related to the ongoing sub-activity.
   - Distinguish between atomic actions, which are short actions, and relationships, which are long-lasting states.

4. RELATIONSHIP
   - Annotate only the relationships that are related to the ongoing actions
   - Group relationships into higher order relationships when they show similar intentions from the actor.

## A2    Data Statistics and Analysis

### A2.1 Class distribution

The section 3 of the main paper has introduced the basic statistics about that MOMA dataset. Here, Figure 2 visualizes the log distribution of character, object, and relationship classes in the data set. It also shows that some characters (e.g., athletes, customer), objects (e.g., chair) and relationships (e.g., holding, on the side of) occur most frequently while other characters (e.g., cashier, driver), objects (e.g., motorcycle) and relationships (e.g., leaning on) have fewer occurrences. For reference, Figure 8, 9, 10 provide the visualization of distributions for activity, sub-activity, and atomic action instances in the MOMA dataset as well.

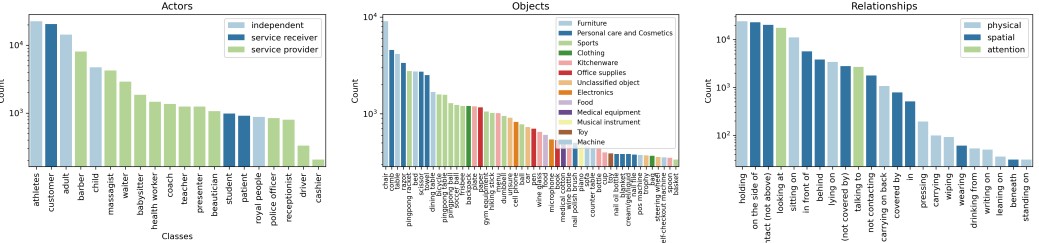

Figure 2: Distribution of (a) character classes, (b) top 50 object classes, and (c) relationship classes.

### A2.2 Class co-occurrence

Further more, we visualize a connection of labels in different abstraction levels using three bipartites, Figure 5, 6, and 7 respectively show a mapping from activity to sub-activity, sub-activity to atomic action, and relationship to entities. In this set of figures, we visualize the distribution of all action class labels and their correspondence to the class labels in the next sub-level.

### A2.3 Action Hypergraph Full Example

In Figure 4, we show full examples of action hypergraph annotated for a video. Notice that the action hypergraph representation changes dynamically as the interactions among entities develop throughout the activity. One goal of activity parsing is to capture the change of the action hypergraph structure to model human's cognitive understanding to the prototypical components of human activities. In this way, we expect model to acquire a more sophisticated cognitive understanding of the on-going activity, and achieve a superior performance on the task of activity parsing.

### A2.4 Significance of Higher-order Relationships

We had a brief discussion in the section 3.3 of the main paper about the significance of higher-order relationships, noting that a regular graph could be converted to a hypergraph by merging edges that have same attributes and come from same nodes. Namely, the scene graph defined in Action Genome [6] or Visual Genome [8] can be converted to hypergraph as well. Therefore, it is easy to develop a false sense of intuition that annotating higher-order relationships is equivalent to annotating its decomposed pair-wise equivalents.

However, we've argued that this is not the case in annotating human interactions. To further illustrate this idea, in Figure 3, we show the cases with trivial merging of pair-wise edges are not equivalent to the annotated higher-order relationships. For example, in the second frame, annotator separated $(C)$, looking at, $(3)$ from $(A, B)$, looking at, $(3)$, as the relationship of looking at reflects different intentions, as the waiter is pouring wine into the glass whereas customers are both in an idle and passive state.

We observed this pattern frequently during annotation procedure and hypothesized that the choice of whether using higher order relationships better reflects human's cognitive understanding to the scene components based on the action. We verify this assumption by showing that preserving the hypergraph structure yields superior recognition performance comparing to decomposing it to a regular graph.

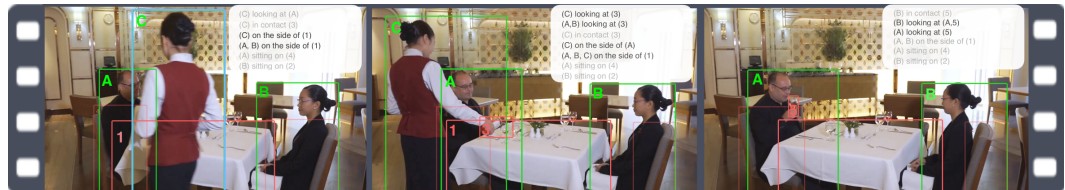

Figure 3: This figure shows high-order relationships cannot be trivially merged from pair-wise ones. Annotated higher order relationships preserve information about human's perception to group interactions when observing an action.

## A3 Experiment Details

As explained in detail in section 5 of the main paper, we assume that the bounding box tracklets for actors and objects are available a priori for all three major experiments we conduct. In this section, we hope to clarify furthe details about our experiment setup, highlighting the information required to reproduce our reported experiment result.

### A3.1 Hypergraph Feature Representation

In this subsection, we introduce the feature representation as well as other details we employed for our `HGAP` and `HGAP` Oracle model reported in section 5 of the main paper.

**Node feature.** We compute a visual feature $\in \mathbf{R}^{512}$. for each bounding box by applying RoIAlign and average pooling on ResNet-18-C4. We then forward the representation to a Multi-Layer Perceptron (MLP) to obtain a node feature $\in \mathbf{R}^{256}$..

**Edge feature.** We extract two types of features for relationships among entities: (1) a vector from the source node to the target node, (2) bounding box coordinates of the associated nodes. We concatenate them into one representation, followed by MLPs, yielding an edge feature $\in \mathbf{R}^{256}$.

**Oracle node feature.** We represent the groundtruth actor or object class as a one-hot vector and encode the vector into a node feature $\in \mathbf{R}^{256}$. using an MLP.

**Oracle edge feature.** Similar to oracle node features, we encode the one-hot vector of groundtruth relationship class into a feature representation $\in \mathbf{R}^{256}$. using an MLP.

### A3.2 Network Architecture and Model Parameters

As shown in Figure 4 of the main paper, the structure of the proposed `HGAP` network consists of a graph stream and a video stream. We train the two streams separately and average the logits in test time. The video stream is a `X3D-L` network [2] that extracts raw video features on a clip level, customized with multi-headed output. The graph stream instead takes in the node and edge features from an entire trimmed video that corresponds to a complete sub-activity. We primarily explain the detail of the graph stream in `HGAP` network below.

**Graph Generation and Graph Encoding module.** We first feed the input into a Graph Generation module which reconstruct the original action hypergraphs, then encode the graphs with a Graph Encoding module. The graph generation module is a three-layer Hypergraph Convolutional Networks (`HGCNs`) [1] with `ReLU` and Batch Normalization [5] between adjacent layers. We then classify the nodes and edges in a graph with their corresponding features. We further encode the predicted network with `HGCNs`, and then concatenate the encoding and the hidden representation from the graph generation module to get the final representation for subsequent tasks.

**Instance module.** The tasks of temporal atomic action localization and role classification are actor-centric. We extract per-instance information by collating node attributes from the same actor across the video. Each per-instance representation preserves the temporal information in a frame-by-frame manner. For role classification, we further apply an average pooling across frames to get a single representation for each actor, before feeding it to an MLP.

**Temporal module.**  For activity classification, sub-activity classification, and multi-label atomic action classification, we first feed the representation to a Temporal module which aggregate temporal information with an average pooling across frames. We then forward the aggregated representation to an MLP to obtain the final prediction.

### A3.3 Training/Optimization Techniques

We implemented our baseline and proposed models with PyTorch and PyTorch Geometric [3]. Our main network is trained with Adam [7] with an initial learning rate of $5 \times 10^{-3}$ and a cosine annealing schedule [9]. We train the network for 100 epochs with a batch size of 32 on the Amazon Web Service (AWS) cluster with 8 Tesla V100 GPUs.

## A4   Future Steps

With the comprehensive annotation in multiple hierarchies of action events, MOMA dataset opens up a wide range of potential future research directions. As examples, this could include (1) action hypergraph generation from video, (2) hierarchical action localization for sub-activity and atomic actions, (3) actor-centric role classification, (4) loss function design for activity parsing model, (5) explainable model visualization with action hypergraph, and many more.

On the other hand, we are planning to update and maintain the MOMA dataset for the long term. In the next few months, we will expand the dataset by at least five times of the current size, making MOMA comparable to Charades [10] (or Action Genome [6]) in size while providing same level quality of fine-grained annotations on each hierarchy of abstraction. We will also provide the dataset API and launch a challenge on Activity Parsing in future conferences. The currently proposed MOMA dataset, APIs, and code implementation for example usages will be released at https://blinded.

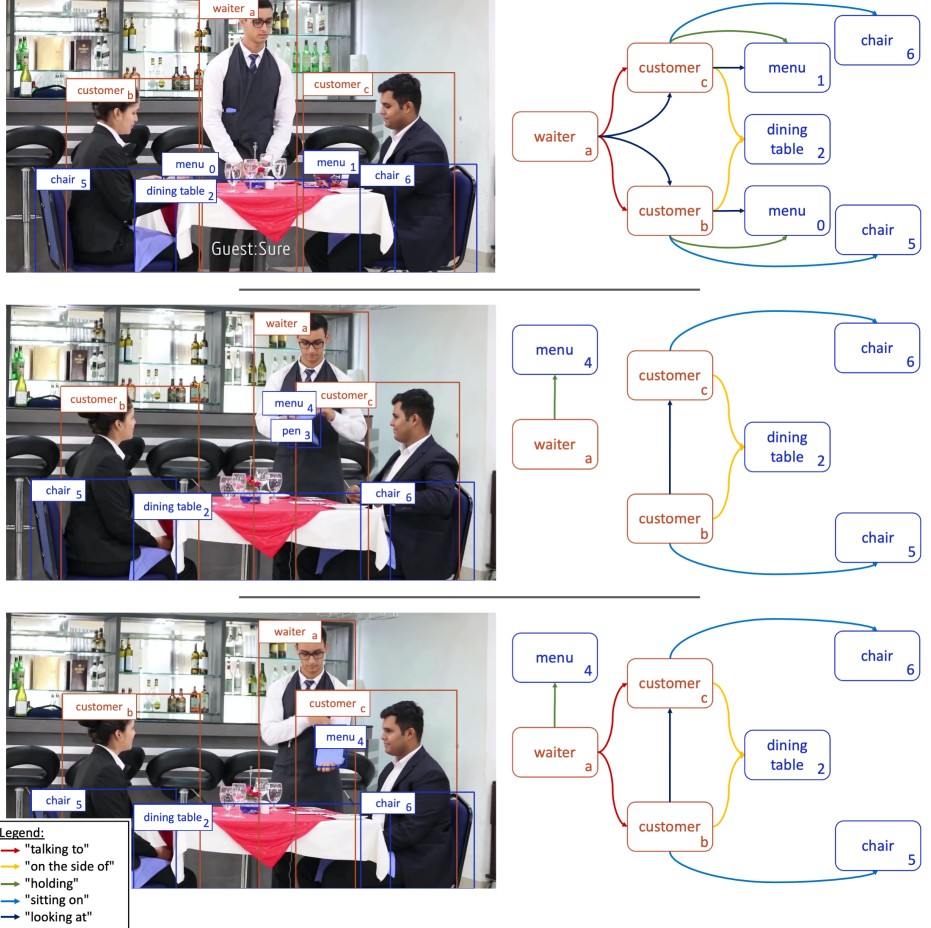

Figure 4: An example of action hypergraph annotation for "take an order" sub-activity in the "dining service" activity. The graph contains entities ("customer", "waiter", "menu", "dining table", etc.) that are localized in each image as bounding boxes. Each entity is bound to a unique identifier which is persistent across frames in the video: actors are associated with alphabetical identifiers, while objects are associated with numeric identifiers. The actors and objects are connected by pair-wise and higher order relationships ("on the side of", "taking to", etc).

Table 1: The taxonomy of actor roles.

| Service provider | Independent | Service receiver |
|---|---|---|
| barber | athlete | customer |
| massagist | royal people | child |
| waiter | adult | student |
| babysitter | | patient |
| health worker | | |
| teacher | | |
| presenter | | |
| beautician | | |
| police officer | | |
| receptionist | | |
| driver | | |
| cashier | | |
| coach | | |

Table 2: The taxonomy of relationships.

| Attention | Physical | Spatial |
|---|---|---|
| looking at | carry on back | above (not covered by) |
| talking to | carrying | behind |
| | drinking from | beneath |
| | holding | covered by |
| | leaning on | in |
| | lying on | in contact (not above) |
| | pressing | in front of |
| | sitting on | not contacting |
| | standing on | on the side of |
| | wiping | wearing |
| | writing on | |

Table 3: The taxonomy of activity.

| Activity | |
|---|---|
| dining service | security screening |
| hospital service | hotel service |
| playing frisbee | making a transaction |
| making a presentation | salon and spa service |
| barber service | babysitting |
| play boardgame | make a western marriage proposal |
| instructing student(s) | group excercise |
| playing ball games | teach riding a bike |
| | coronation |

Table 4: The taxonomy of sub-activities.

| Subactivity | |
|---|---|
| serve food and utensils | cut nails |
| clean skin with achohol | teach riding a bike |
| catch frisbee | shoot the gate (in soccer) |
| welcome award winner | play child ball games |
| stop bleeding | nurse someone |
| cut hair | cut beard |
| put on ring (for someone) | perform injection |
| comb hair | group jogging |
| tutoring for coursework | hug each other |
| playing pingpong | perform sacred ritual |
| bag screening for security check | dry hair |
| make a public speech | make a demo |
| clean up table | perform dental examination |
| shave (during shaving beard) | body screening for security check |
| catch the soccer ball | polish nail and skin |
| shave (during haircut) | kiss each other |
| take orders | serve water |
| valet parking | serve wine |
| order drive thru food | hug child |
| give a body massage | take care of child |
| pick up drive thru order | wash hair |
| front-desk reception | perform soberity test |
| give an award speech | help child get on bike |
| play with child | perform eye examination |
| throw frisbee | pass the ball (in soccer) |
| serve the guest | cheers with each other |
| checkout at supermarket | teach playing piano |
| hand over award | fall off from bike |
| apply cream/gel/liquid | gym equipment training |
| give a skin massage | skin care |
| order or pay with machine | make a speech on graduation ceremony |
| play boardgame | apply products |
| group hiking | process (during ritual) |
| | play basketball |

Table 5: The taxonomy of objects categories.

1.2

| Payment | Sports | Kitchenware | Furniture | Performance | Clothing/Wearables | Electronics | Medical Equipment | Personal care and Cosmetics | Office supplies | Vehicle | Others |
|---|---|---|---|---|---|---|---|---|---|---|---|
| pos machine | pingpong racket | plate | chair | piano | backpack | cell phone | medical cotton | comb | paper | car | unsure |
| self-checkout machine | bicycle | menu | table | piano book | blanket | microphone | tweezers | razor | pen | steering wheel | stick |
| card | pingpong table | wine glass | bed | guitar | bag | digital device | syringe | scissor | book | motorcycle | animal |
| order machine | pingpong ball | wine bottle | dining table | trophy | wedding ring box | screen | human model | towel | box | car seat | ritual equipment |
| wallet | soccer ball | bottle | sofa | poker card | hat | telephone | dental equipment | nail polish brush | tray | window | crown |
| goods | frisbee | cup | counter table | podium | clothes | computer | alcohol pad | nail oil bottle | corkscrew | | |
| food | gym equipment | spoon | toy | | diaper | laptop | breathalyzer | cream/gel/liquid | | | |
| cash | hiking stick | bowl | stool/bench | | decoration item | camera | eye examination tools | nail file | | | |
| cart | dumbbell | fork | bench | | wedding ring | | flashlight | hair drier | | | |
| atm | ball | utensil | armrest cushion | | | | | massage chair | | | |
| | basket | baby feeder bottle | board | | | | | massage table | | | |
| | soccer gate | knife | pillow | | | | | laying chair | | | |
| | basketball | kettle | child's chair | | | | | clip | | | |
| | | sink | carpet | | | | | mirror | | | |
| | | | cabinet | | | | | bath bucket | | | |
| | | | | | | | | nail clippers | | | |
| | | | | | | | | toe separator | | | |
| | | | | | | | | tooth brush | | | |
| | | | | | | | | brush | | | |
| | | | | | | | | sponge | | | |

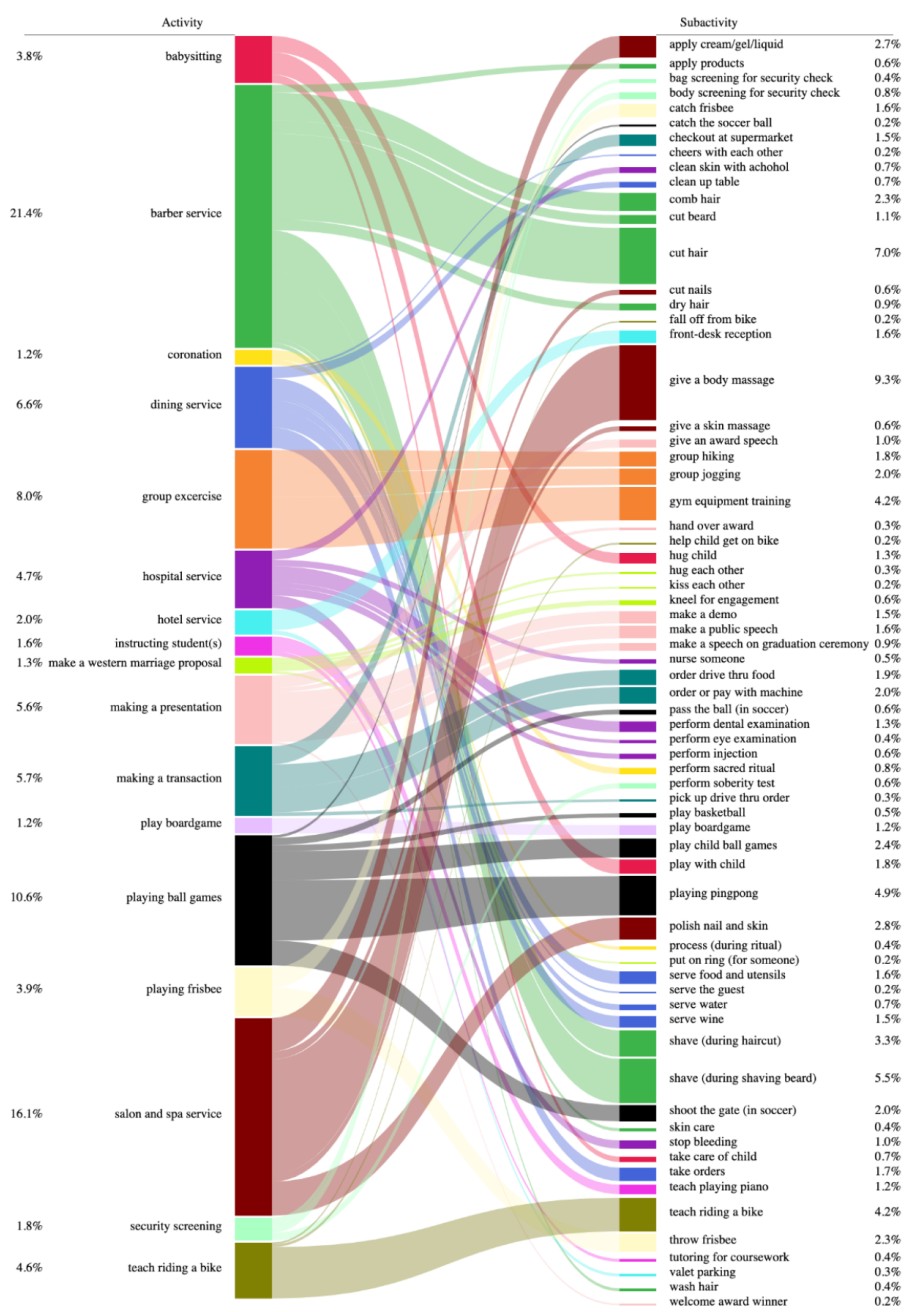

Figure 5: A weighted bipartite mapping from activity to sub-activity. We show a surjective mapping for sub-activities to activities so that each sub-activity only belongs to one activity.

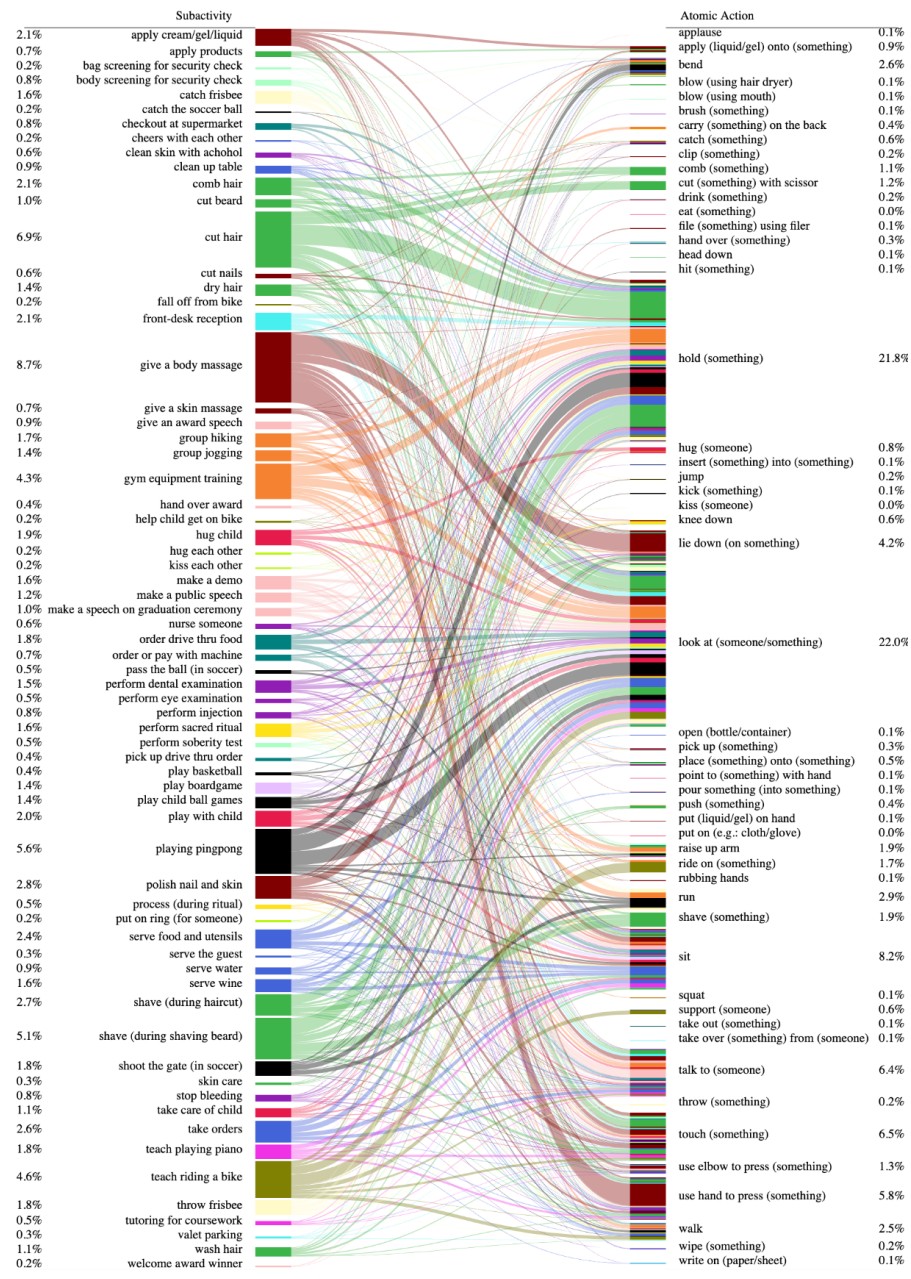

Figure 6: A weighted bipartite mapping between sub-activity and atomic actions. This demonstrate that same atomic actions can occur in different sub-activities.

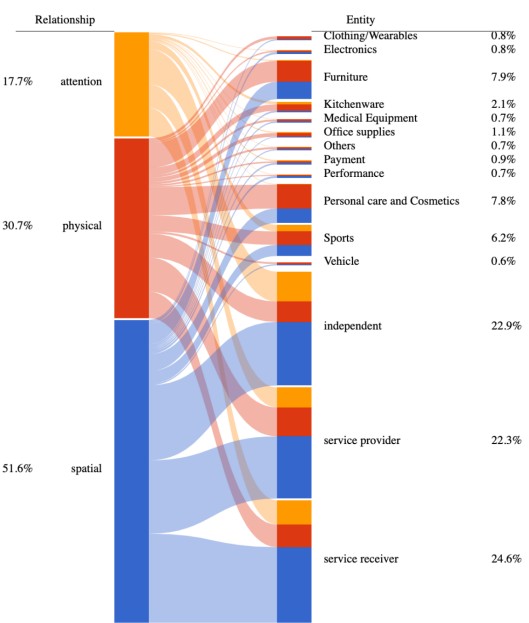

Figure 7: A weighted bipartite mapping between high-level relationship categories and high-level actor and object categories shows that they are densely connected.

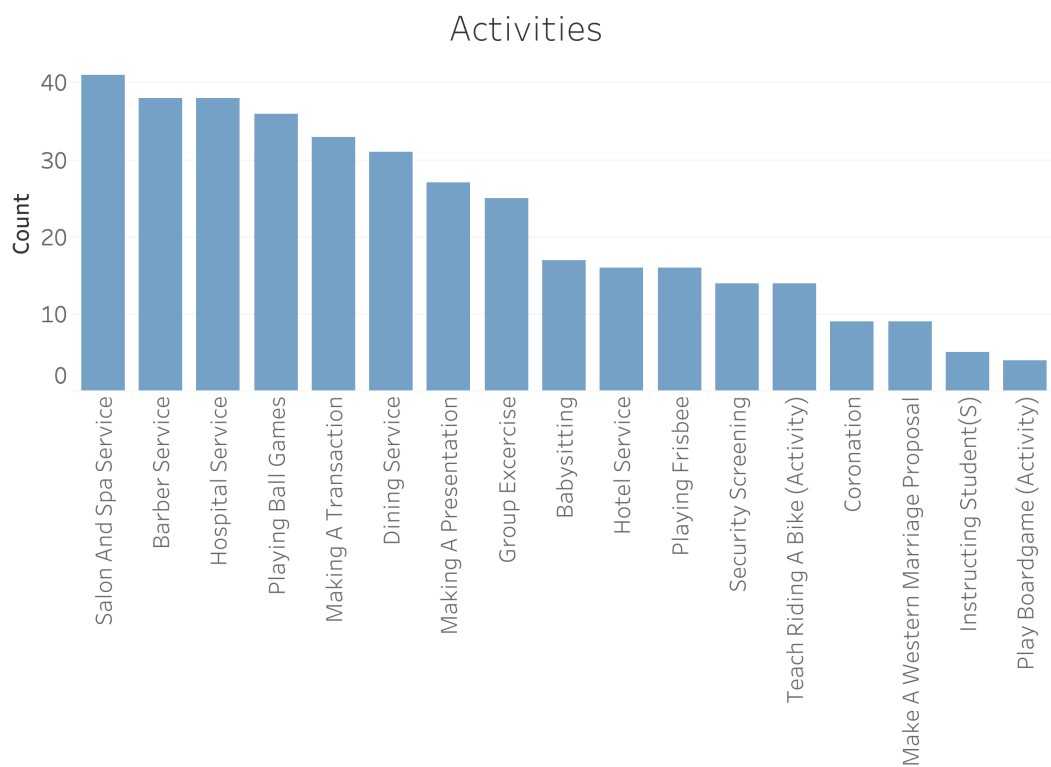

Figure 8: Distribution of activities. Each untrimmed video contains exactly one activity and we have in total 373 untrimmed videos, each of which can last between 10 minutes to an hour.

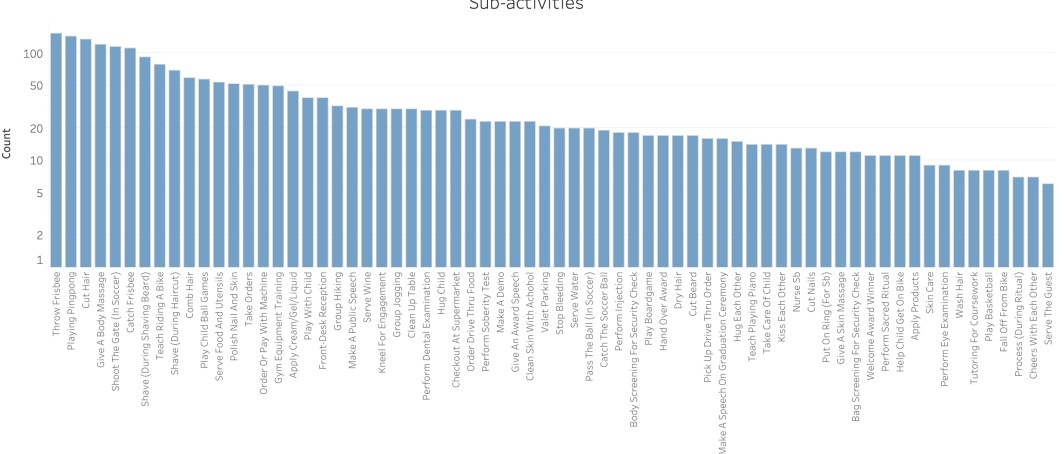

Figure 9: Distribution of sub-activities. Each trimmed video corresponds to exactly one sub-activity, and is created based on the temporal coordinates of the sub-activity in the raw video. We have a total of 2,364 trimmed videos, each of which can last between several seconds to 3 minutes.

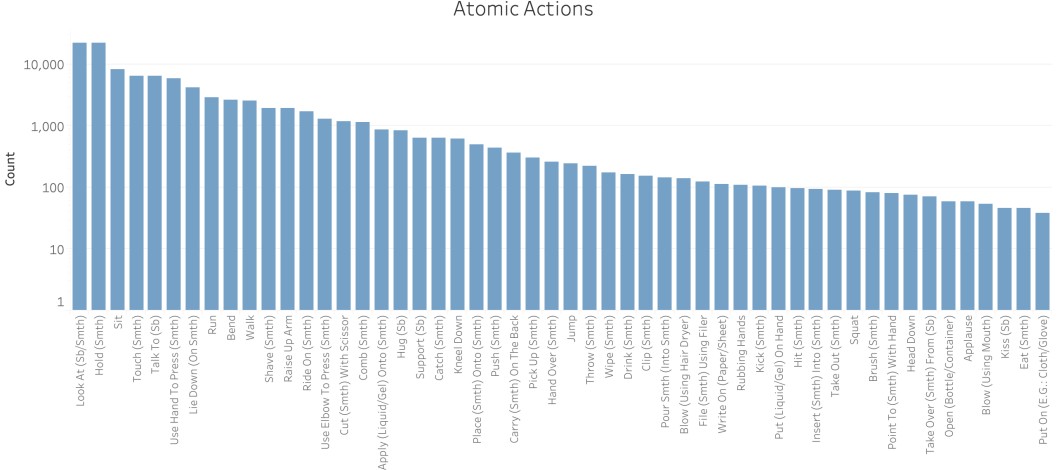

Figure 10: Distribution of atomic actions, each atomic action is localized to the corresponding actor. We have a total of 12K instances of atomic actions, and a total of 100K localized atomic actions across all the frames in the dataset.