# OpenReview forum: "MOMA: Multi-Object Multi-Actor Activity Parsing"
_NeurIPS.cc/2021/Conference — NeurIPS 2021 Poster_

### Official Review · Reviewer_hk9t · 2021-07-11

**Rating:** 7
**Confidence:** 5

**Summary:**

The paper introduces a new dataset for video-based activity parsing. Instead of assigning a single action label to a video, the paper decomposes an activity into sub-activities and atomic actions. Instance-level annotations of actors, objects and their relationships are also annotated. Based on this dataset, the paper proposes a new model, action hypergraph, for activity parsing. Experimental results demonstrate the potential applications of the dataset and the effectiveness of the proposed method.

**Limitations And Societal Impact:**

Limitations and potential negative societal impact are well addressed.

**Main Review:**

Strengths

+ The task of activity parsing is important and is likely to be the next step of video understanding beyond the current popular tasks of action recognition and detection.
+ The dataset is new and well designed. It is a good supplement to existing datasets, mostly on action recognition and localization, and will benefit the computer vision community.
+ The paper tries to give definitions of the subjects of study (activity, sub-activity, atomic action).
+ The use of hyper-graphs to solve this problem is new and makes sense.
+ Several tasks are defined and evaluated on this new dataset.

Weaknesses

- Parsing an activity into a hierarchy of sub-activities and atomic actions with spatial-temporal human-object interactions is not new. Several prior works on this direction are ignored. The authors need to credit those pioneering works and compare with them on the task and datasets.

[1*] Wei, Ping, Yibiao Zhao, Nanning Zheng, and Song-Chun Zhu. "Modeling 4d human-object interactions for event and object recognition." ICCV 2013.

[2*] Pei, Mingtao, Yunde Jia, and Song-Chun Zhu. "Parsing video events with goal inference and intent prediction." ICCV 2011.

[3*] Wei, Ping, Yang Liu, Tianmin Shu, Nanning Zheng, and Song-Chun Zhu. "Where and why are they looking? jointly inferring human attention and intentions in complex tasks." CVPR 2018.

- I feel the tasks in Experiments fail to make full use of the dataset and diverge from 'activity parsing'. The input is a trimmed video with a single sub-activity segmented from the complete video. Why not use an input video at the activity level? If a sub-activity clip is the input, the task should be called sub-activity parsing instead of activity parsing.

- Line 234. It is fine an existing method is used for activity parsing. But, for self-completeness, it is better to briefly introduce what the complex equation, including a lot of matrix multiplications, is doing. Also, H seems not defined.

- Since the relationships among humans and objects are not known in advance, I wonder how the spatial-temporal hyper-graph is constructed. Will considering high-order relationships bring a large number of edges (for example, via enumerating)?

**Time Spent Reviewing:**

4 hours

---

> ### Author Response · Authors · 2021-08-11
> **Thank you for your review**
>
> We thank the reviewers for their positive assessment of our work and helpful suggestions for improvement. We address the reviewer’s comment below:
>
> - **Novelty of hierarchical activity recognition**: We did not claim that we were the first to introduce hierarchical activity recognition. As shown in Figure 1 (left), our work generalizes and complements previous approaches for complex activities that involve multiple humans interacting with various objects. Thank you for bringing our attention to the three papers by Wei et al. and Pei et al., and we will make sure to cite these papers accordingly.
> - **Inputs are trimmed videos**: For the ‘activity parsing’ defined in this paper, we didn’t include temporal action localization for sub-activity as part of the parsing task, so we need to take in trimmed videos to predict results for all levels at the same time. This is because, in the scope of this paper, our main effort is to demonstrate the effectiveness of the action hypergraph definition, instead of designing the activity parsing to incorporate every possible video understanding task, even though it is the far reaching goal we hope MOMA dataset can bring the community one step closer to. So far, we have managed to expand the dataset 5x during the review process and the latest dataset contains 1416 untrimmed videos and 15,876 trimmed videos. This makes a richer and more comprehensive definition of activity parsing possible in the next step of research.
> - **Explain Hypergraph neural networks**: The explanation and derivation of the equation in line 234 can be found in [18]. “H” refers to the hypergraph adjacency matrix (also known as incidence matrix) as explained in [18]. Because of limited space, we decided not to elaborate this equation further.
> - **How to construct relationships**: As the reviewers pointed out, listing every possible pair relationship is intractable. While trying our best to be comprehensive, we only annotate and expect the model to recognize relationships that are relevant to the higher level activities and sub-activities.

---

> > ### Comment · Reviewer_hk9t · 2021-08-31
> > **after rebuttal**
> >
> > After going through all reviews and authors' responses, I keep my positive rating.

---

### Official Review · Reviewer_PT1W · 2021-07-13

**Rating:** 7
**Confidence:** 4

**Summary:**

This paper proposes that video understanding be approached by trying to recognize at several different levels of granularity, simultaneously. At the highest temporal granularity is the "activity", which consists of a sequence of temporally-localized "sub-activities". Below that are the "atomic actions" performed by individual actors, and at the bottom an "action hypergraph" that is a scene hypergraph showing the relationships between actors & objects at the frame level.
The authors introduce a video dataset, MOMA, and a task "Activity Parsing" which aims to infer all of these annotations from a video together with ground truth person and object tracks. Further they propose a HyperGraph Action Parsing Network, and compare it to 3DCNNs and graph convolutional networks on the Activity Parsing task.


**Ethical Concerns:**

This paper doesn't investigate or address potential biases in the dataset (e.g. gender, race, age), which I think could crop up if anyone tried to apply a model trained on this benchmark to a real-world scenario. The authors could add a disclaimer accordingly.

**Limitations And Societal Impact:**

Not addressed. Any video action understanding model can be applied to the task of surveillance, although the labels introduced in this dataset seem benign.

**Main Review:**

[Originality] Although there have been a number of recent proposals formulating "temporal scene graphs for video", this paper provides a new perspective that I think is valuable, in particular because it unifies several video tasks -- classification, action localization, trimmed clip classification, spatiotemporal atomic actions, and visual relationships -- all under a single umbrella and embodied in a single dataset.

[Quality] The paper is generally very good. In particular the dataset represents a significant investment and seems valuable to the community. The HGAP model seems not particularly compelling due to uneven performance on the task, but that's fine for a dataset baseline. However my main question is around the ability to infer hypergraphs from video. For the technique to be useful, ultimately they will need to be inferred from raw, unannotated video, but in the paper's experiments, the graphs are given as input. Realizing the many degrees of freedom a hypergraph has, how reasonable is it that we can expect a model to be able to infer them from raw data? If inferring reasonable hypergraphs is very hard, then the modeling approach falls down. It would be nice to see how sensitive the approach is to errors in inferred hypergraphs; initial results in Table 2 (Oracle vs. non-Oracle) suggest that it might be very sensitive.

Similarly, it would be nice to see how reliably human annotators can produce a consistent, agreed-upon hypergraph for a video. This would be a good upper-bound for how well we think a machine could do it -- if people can't agree on the correct hypergraph, how could a machine do it?

[Clarity] The paper is clearly written, and generally well organized. However it's a little confusing that the experimental section is divided into three tasks. Upon digging deeper it's apparent that two of the tasks--5.2 Hierarchical Action Classification and 5.3 Role classification--are simply subtasks of 5.1 Activity Parsing, so it could just be a single task.

Another point that could be more clear is that the actual hypergraph structures are provided as input to HGAP / GCN, and not inferred directly from the input videos. I realized this only after seeing "action hypergraph generation from video" listed as future work in the appendix.

[Significance] I think the dataset is particularly valuable since connects coarse activities, to atomic actions, and to video scene graphs, which could enable research into more "holisitic" video understanding methods. Although its size is currently moderate, the authors note that it will soon contain 5x more annotated data, which will be impressive. Judged mainly on the significance of the dataset, I think this paper is valuable.


**Time Spent Reviewing:**

3

---

> ### Author Response · Authors · 2021-08-11
> **Thank you for your review**
>
> Thank you very much for all the comments. We are glad that you think our dataset is ”particularly valuable”. Please see our responses below.
>
> - **Inferring from raw videos**: We think it's reasonable to assume object boundings in activity parsing since object detection is a separate and well-studied problem, and we want to focus on temporal reasoning. However, we do strongly agree that performing activity parsing from directly raw videos is an important next step for this project. Our group is actively developing new activity parsing models based on the MOMA dataset and inferring action graphs from raw videos is one of our major goals. Our hope is that the comprehensive annotations in the MOMA dataset will inspire new activity parsing models that have more relaxed assumptions.
> - **Annotation consistency**: This is a common problem which also exists in object detection and scene graph generation datasets. While it is difficult to address this concern completely, our co-authors have the past experience of collecting one of the largest and most widely used datasets on visual relationships. We tried our best to derive a set of rules and heuristics based on past experience and we will provide the details in the appendix.
> - **Breakdown of tasks**: We cannot trivially merge the three tables because (1) these three tasks are intrinsically different and (2) it makes comparing with existing baselines easier and more straightforward. Activity Parsing (5.1) focuses on temporal action localization between activity&sub-activity and between sub-activity&atomic action. Hierarchical Action Classification (5.2) considers multi-label classification at the sub-activity level for easy comparison with existing activity classification models. Lastly, Role Classification (5.3) is a task that is rarely discussed in existing work but critical to understanding video semantics.
> - **Ethical concerns**: Thank you for pointing out the potential ethical concerns. We will follow the suggestion and add a disclaimer accordingly. We are planning to further sanitize the dataset in the future.

---

### Official Review · Reviewer_QrDE · 2021-07-19

**Rating:** 7
**Confidence:** 2

**Summary:**

This paper introduces Activity Parsing for the understanding of complex human activities based on a hierarchical structure describing human actions at 4 different levels. As an improvement from traditional pairwise entity representations, the Action Hyperedge is used to represent the more complex relationships between multiple humans and objects. Also the Action Hypergraph provides better interpretability for reasoning of high-level actions or activities. Followed by the concept of the hierarchical Activity Parsing, a new, meticulously-annotated video dataset, MOMA, is released and an HGAP network is proposed. In the experiment, the HGAP network achieves comparable results with the state-of-the-arts in several tasks such as video and role classification.


**Ethical Concerns:**

The MOMA dataset is collected and annotated from public videos. The authors make a large effort to mitigate the ethical concerns in section 3.1. The dataset is not accessible during the review process, and careful checks should be carried out before the release.


**Limitations And Societal Impact:**

See above for discussion on limitations of work.

The authors should elaborate on the discussions of the benefits/tradeoffs of the HGAP network.



**Main Review:**

This paper proposes a novel hierarchy for parsing human activities, annotates a dataset based on the concept, and builds a network for activity parsing, and role/object classification. Works including video datasets for human activity recognition, action parsing methods, object detection, and graph neural networks are adequately cited.

It resolves the limitation in prior works in understanding complex, multi-person, multi-object human activities, and points out a new direction for video parsing. The release of the dataset will surely benefit the research in video understanding and human action detection/recognition. The proposed HGAP network takes advantage of the activity structure and combines the techniques in graph neural networks with video encoding networks for activity parsing and some downstream related tasks.

Technical soundness:

The design of the activity hierarchy and the HGAP network is quite reasonable. The authors consider the activity parsing as a set of several graph/edge/node classification problems and show in the experiments that the HGAP network outperforms the baselines on high-level tasks. However, at graph-level tasks, GCN outscores HGAP by a larger margin (Table 2). It’s not accurate to claim HGAP outperforms GCN in the abstract.
It is also surprising that GCN even outperforms HGAP Oracle at graph-level tasks. Ideally the ground truth hypergraph can be decomposed into pairwise graphs, so HGAP Oracle should be easier to train than GCN for both graph-level and high-level tasks. Please provide further explanations and deeper discussions for this.

Clarity of Expositon:

Overall the clarity for the activity hierarchy and the dataset is good. Clear examples and illustrations (ex. the dining service) are used for explanation. However, there are some remaining questions.
1. I agree the design of hyperedges is necessary because they better represent high-order actions, but the explanation for your hypergraph over the merged hypergraph from pairwise edges is not clear. Is this reason correct -- the two connections, (waiter, looking at, glass) and (customers, looking at, glass) cannot be merged because they might belong to different sub-activities? Please explain.
2. It’s not very clear why the HGAP network is designed with several readouts until readers go through section 5. Some readouts are only used for specific tasks, so it would be better if illustrations of how the different levels of readouts are connected to the tasks in 5.1 can be shown in Figure 4.
3. It remains unclear where the hypergraphs come from in the training process, since some components of them are the outputs of some tasks (ex. edge classification). The HGAP Oracle setup uses the ground truth graph structure. What about HGAP? Is the graph structure learned in the training process? If yes, how does it influence the classification results?
Also, are different weights used for different edge types in graph neural operations?

Some minor errors:
1. Figure 3 is not mentioned in the paragraphs.
2. In line 209, it should be to^(no).
3. In line 237, the term ROI shows up for the first time. Please provide its full form.




**Time Spent Reviewing:**

2-3 hours

---

> ### Author Response · Authors · 2021-08-11
> **Thank you for your review**
>
> Thank you for your feedback! It is truly appreciated that you provided detailed editorial comments and suggestions that improve the clarity of the paper.
>
> - **GCN outscores HGAP**: HGAP outscores GCN on four metrics at activity, sub-activity, and atomic action levels. At the action hypergraph level, GCN indeed outperforms HGAP on role classification. This aligns with our intuition that hyperedges and hypergraph convolutions are crucial for high-level activity understanding (and thus the performance gain), but not so important for role classification. GCN indeed has a better inductive bias for certain tasks (such as role classification) and the next step of this project is to combine the strengths of both methods.
> - **Why is hyperedge better**: We appreciate the insightful question. To clarify, the two relationships may belong to different atomic actions (not sub-activities). Such observation can only be determined based on human’s understanding of the ongoing event. In Supplementary Material section A2.4 and Supp. Fig. 3, we have provided a detailed explanation and visualization to the question of why certain pair-wise relationships cannot be trivially merged.
> - **Readouts of HGAP**: Thank you for the editorial comments. We will update the draft accordingly to connect Section 5.1 and Figure 4.
> - **Details of hypergraphs**: As explained in line 272-280, HGAP assumes bounding box tracklets (without knowing object classes) and HGAP oracle assumes ground-truth action hypergraph structures (including bounding box tracklets and edge connections, without knowing object and edge classes). HGAP oracle establishes an upper bound of HGAP on activity parsing. Edge weights are not manually set but are implicitly inferred in graph convolutions.

---

### Official Review · Reviewer_Etdz · 2021-08-02

**Rating:** 6
**Confidence:** 4

**Summary:**

The paper defined a new action representation called action hypergraph which connects multiple entities with hyperedges, the role classification for actors, and the  temporal tracking for entities. A new dataset (MOMA) following the defined representation is proposed. The paper also proposed a HyperGraph Activity Parsing (HGAP) network for tackling multiple activity understanding tasks simultaneously, e.g. activity recognition, sub-activity recognition, atomic action recognition, etc. Results for these activity recognition tasks, as well as the hypergraph construction evaluation results (hyperedge classification and role classification) are reported on MOMA, together with some simple baseline results.


**Limitations And Societal Impact:**

More extensive evaluation should be conducted to tackle the concern in the above "Main Review" section.


**Main Review:**

1. The definition of action hypergraph is technically sound, since each action may involve multiple objects, and the action role and temporal tracking are also important for action understanding. The paper also proposes a new multi-task framework "HGAP network", and reports the results for each task separately. However, the results on Table 2 are hard to differentiate whether the result improvement comes from the action hypergraph definition with hyperedge, or is due to the proposed multi-task model "HGAP network". The baseline "GCN [31]" is modified from the Graph Convolutional Network (GCN) model [31] where hyperedges are decomposed to pair-wise ones. It is better to change hyperedge to pairwise edge connection in "HGAP network" to show the advantage of action hypergraph definition.

2. All the results in this paper are built on top of GT bounding boxes (without class labels) and GT tracklets. Results using predicted bounding box proposals instead of GT bounding boxes should be reported, as is common practice in relevant tasks, e.g. dense captioning, scene graph generation, etc.

3. For the role classification performance of proposed HGAP network in Table 4, it is compared with the baseline model ResNet-50 (37.9% mAP). However, another baseline model "GCN [31]" in Table2 reports a role classification accuracy of 64.7% mAP, which is significantly higher than both the baseline ResNet-50 (37.9% mAP) and proposed HGAP model (46.4% mAP) results on Table 4. Though the author has some explanation for this low baseline result in Table 4, but the explanation is not convincing. ".........To emphasize the importance of visual relationships, this baseline does not use graph structure on purpose.........."


**Time Spent Reviewing:**

3

---

> ### Author Response · Authors · 2021-08-11
> **Thank you for your review**
>
> We thank the reviewers for their assessment and constructive suggestions. We individually address questions and concerns below.
>
> - **Interpreting Table 2**: We agree and are fully aware that it is better to change hyperedge to pairwise edge in “HGAP” to show the advantage of action hypergraph. This is why in Table 3, we added the “X3D+GCN” model, which has exactly the same structure as HGAP except that we replace hypergraph convolution with graph convolution on decomposed pairwise edges. We can still see result improvement for the HGAP when compared to the “X3D+GCN” model, which shows the significance of action hypergraph definition. We will make sure to include this result in Table 2 as well.
> - **Results using bounding box proposals**: This paper does not concentrate on static scene understanding because other studies on dense captioning or scene graph generation have demonstrated promising results. Instead, this paper focuses specifically on temporal reasoning. Meanwhile, our group is working on a separate project that will develop an end-to-end activity parsing system using raw untrimmed video as input. We hope the comprehensive annotations in the MOMA dataset will inspire new activity parsing models with more relaxed assumptions.
> - **Role classification results**: This concern is in fact addressed in the “Results” section 5.1 for Table 2, so it is not repeated in the role classification section. To rephrase, we believe learning hypergraphs is more challenging than learning regular graphs, which is why we see inferior results on edge and role classification even if the Oracle hypergraph is given. However, the very fact that HGAP still performs better on all high-level tasks in turn testifies that action hypergraphs can provide richer information even if they are not fully learned. We anticipate further improvements on high-level recognition tasks if a more capable hypergraph learning model can be designed in the future.

---

> > ### Comment · Reviewer_Etdz · 2021-08-30
> > **after rebuttal**
> >
> > The author's rebuttal gave answers to my listed three questions. For the last two questions, using GT boxes instead of learning from raw videos, and inferior "Role classification" results in Table 4, the authors gave their justifications as follows.
> >
> > “Results using bounding box proposals: This paper does not concentrate on static scene understanding........... Instead, this paper focuses specifically on temporal reasoning......."
> > "Role classification results:.......However, the very fact that HGAP still performs better on all high-level tasks in turn testifies that action hypergraphs can provide richer information even if they are not fully learned....." This means that, though their "Role classification" results are not so good, the high level tasks built on top of the imperfect representation can still achieve better performance.
> >
> > I could also accept these two justifications, but I also hope that, these justifications could be equally accepted for other papers to justify some of their imperfect results in their papers, and become standard practices.

---

### Decision · Program_Chairs · 2021-09-27

**Decision:**

Accept (Poster)

**Comment:**

The authors introduce action hypergraph and an associated task (activity parsing), as well as a new dataset (MOMA) for complex activity recognition.  All of the reviewers are positive, and find the representation has certain novelty and the dataset will be valuable to the community.  The AC concurs that this paper is acceptable, and recommends a poster.